# Development and validation of the Iranian Minimum Data Set for Epidermolysis Bullosa: A mixed method approach

**Somayeh Paydar[1], Shahrbanoo Pahlevanynejad[2], Farkhondeh Asadi[3], Hamideh Ehtesham[4], Azam Sabahi[4]***

**1** Assistant Professor of Health Information Management, Department of Health Information Technology, School of Allied Medical Sciences, Kermanshah University of Medical Sciences, Kermanshah, Iran, **2** Assistant Professor of Health Information Management, Department of Health Information Technology, School of Allied Medical Sciences, Semnan University of Medical Sciences, Semnan, Iran, **3** Professor of Health Information Management, Department of Health Information Technology and Management, School of Allied Medical Sciences, Shahid Beheshti University of Medical Sciences, Tehran, Iran, **4** Assistant Professor of Health Information Management, Department of Health Information Technology, Ferdows Faculty of Medical Sciences, Birjand University of Medical Sciences, Birjand, Iran

* sabahiazam858@gmail.com

**Data Availability Statement:** All relevant data are within the manuscript and its Supporting Information files.

## Abstract

Minimum Data Set (MDS) enables integration in data collection, uniform data reporting, and data exchange across clinical and research information systems. The current study was conducted to determine a comprehensive national MDS for the Epidermolysis Bullosa (EB) information management system in Iran. This cross-sectional descriptive study consists of three steps: systematic review, focus group discussion, and the Delphi technique. A systematic review was conducted using relevant databases. Then, a focus group discussion was held to determine the extracted data elements with the help of contributing multidisciplinary experts. Finally, MDSs were selected through the Delphi technique in two rounds. The collected data were analyzed using Microsoft Excel 2019. In total, 103 data elements were included in the Delphi survey. The data elements, based on the experts' opinions, were classified into two main categories: administrative data and clinical data. The final categories of data elements consisted of 11 administrative items and 92 clinical items. The national MDS, as the core of the EB surveillance program, is essential for enabling appropriate and informed decisions by healthcare policymakers, physicians, and healthcare providers. In this study, a MDS was developed and internally validated for EB. This research generated new knowledge to enable healthcare professionals to collect relevant and meaningful data for use. The use of this standardized approach can help benchmark clinical practice and target improvements worldwide.

## 1. Introduction

Epidermolysis Bullosa (EB) is a group of rare diseases that cause blisters to form in the area of the skin between the outer and middle layers, known as the basement membrane. This disease

**Funding:** The author(s) received no specific funding for this work.

**Competing interests:** The authors have declared that no competing interests exist.

makes the skin and mucous membranes fragile and prone to blistering. There are different types of EB, depending on which layer of the skin is affected and how severe the symptoms are infections, hair loss, and breathing problems can also occur [1]. In severe cases, a child with EB often dies during the first year of life. [2, 3]. This disease is caused by pathogenic mutations in 20 distinct genes that affect cellular integrity and adhesion [4]. The diagnosis of EB requires correlating clinical, electron microscopic, and immunobiological features with mutational analyses, and treatment aims to minimize blistering, provide wound care, relieve symptoms, and address specific complications [5]. It affects all racial and ethnic groups without a specific gender predominance [6, 7]. Symptoms usually appear at or near birth and persist throughout life [8, 9]. However, the onset of lesions in some individuals may not occur until adolescence or early adulthood, and in certain types of EB, blisters may improve with age [10, 11]. EB affects 1 out of every 20,000 births in the United States, meaning approximately 200 children are born with EB each year [12]. According to the latest reports, the prevalence and incidence rates (per million) are 22.4 and 41.3 in the Netherlands, and 11.1 and 19.6 in the United States, respectively [13]. The prevalence rates (per million) are 10.3 in Australia, 19.5 in New Zealand, and 6.7 in Iran [14, 15].

This disease affects about half a million people worldwide, many of whom are children [16]. According to the National Registry of EB patients, 50 out of every 1 million live births in the United States are affected by this disease [17]. Children with EB can experience extensive and complex problems such as persistent itching, frequent painful grooming, and fragile skin [18]. They may also face complications such as anemia, malnutrition, growth delays, deformities, scarring, skin cancer, and eye problems. People with EB may encounter emotional and mental challenges such as depression, anxiety, low self-esteem, isolation, and stigma [19]. They may also have difficulties in school, work, relationships, and hobbies due to their physical limitations and appearance [20]. Individuals with EB require frequent and specialized medical care to prevent and treat their skin problems. They may need to use various products such as bandages, dressings, creams, ointments, and supplements to protect and heal their skin. Additionally, they may need to undergo surgeries or procedures such as skin grafts or gene therapy to improve their condition. Consequently, the cost of care for people with EB can be very high and may not be covered by insurance or public health systems [17].

In a study conducted in 2021 at the Burn Hospital of Tehran University of Medical Sciences, 538 patients with bullous dermatitis were registered in the Iranian EB registry between January 2017 and September 2017. The youngest patient was 10 days old, while the oldest patient was 64 years old. Among them, the dystrophic type (75.7%) was the most common. Most EB patients were of school age. Regarding the family relationships of the patients, all patients with the bonded type had a family relationship between their parents. However, this rate was 84.4% for dystrophic EB, 50% for simplex EB, and 67% for Kindler-type EB. The patients also exhibited varying degrees of dysphagia, dental disease, contractures in the upper and lower limbs and fingers, ophthalmological issues, seizures, and mental disorders. Accordingly, the prevalence of EB in Iran is lower than in other countries, which is likely to increase with improved screening services. Most patients lived in the four provinces of Tehran, Isfahan, Khuzestan, and Fars, making it essential to provide facilities for patients in these regions. Additionally, the dystrophic type of EB was more common in Iran than in other countries [15]. For this reason, disease prevention through genetic analysis and testing during pregnancy is of considerable importance. Accurate diagnosis of the disease type using immunofluorescence tests and electron microscopy helps determine the exact prevalence of each type. The number of simplex cases is gradually increasing with the complete registration of almost all patients [21, 22]. The calculation of the economic burden of disease in Iran showed that 12%

of the total costs are related to direct medical costs, 10% to direct non-medical costs, and 78% to indirect costs [23].

According to the World Health Organization, accurate, timely, and accessible information for healthcare providers plays an essential role in planning, developing, and supporting healthcare services. In other words, healthcare providers need accurate and timely information to perform their professional and specialized activities [24]. Therefore, the first step in the information management of a disease is the design and implementation of a minimum data set (MDS) in health and treatment centers, which can improve the quality of care and disease control [25].

Information technology is perceived as an important tool for healthcare organizations and institutions to manage and improve services in the face of increased demand [26, 27].

The MDS is a standardized approach to data collection that provides accurate access to health data. With the development of public health surveillance, it advances systematic collection, interpretation, comparison, and integration of data, providing solutions regarding health-related threats [25]. MDS can provide valuable information for various aspects of EB, such as its prevalence, incidence, distribution, trends, patterns, and risk factors in different populations and regions. It can also enhance the understanding, diagnosis, treatment, and quality of life for people with EB, as well as their families and caregivers [14]. Furthermore, it can facilitate research and development of new therapies and interventions for EB while raising awareness and support for individuals with EB in society [28].

Essential datasets for EB have been developed in some countries around the world, but the existence of certain differences highlights the need to create indigenous datasets for disease management. Elements such as nationality, residency status, race and culture, and medical and family history are determined based on the indigenous needs of each country and are among the most important differentiators in various MDSs [29]. Many international efforts are underway to develop MDS in most countries, and these initiatives are designed to improve the comparability of disease data across national borders [30].

From 1986 to 2002, the National Epidermolysis Bullosa Registry (NEBR) in the USA was a major source of epidemiological data on EB. The NEBR enrolled more than 3,000 EB patients and collected information on their demographic, clinical, genetic, and quality-of-life characteristics. The NEBR also tracked the prevalence, incidence, natural history, and complications of EB across different types and subtypes of the disease [19, 31]. The international DEB Patient Registry was established to collect and share phenotypic and genotypic data on dystrophic epidermolysis bullosa (DEB) patients from various countries. Akker's study showed that the registry facilitated the diagnosis and genetic counseling of DEB patients while providing novel insights into the rare phenotypes of DEB that are poorly understood [32].

Given that information on the different types of EB has direct clinical applications in terms of revised classification, improved genetic counseling, and the development of DNA-based prenatal testing for families with EB, the need for MDS in this disease is critical [33]. It is necessary to develop MDS to provide the infrastructure for EB disease registration systems, report the incidence and prevalence of the disease, evaluate the quality of care, improve research, and regularly report health information for these patients. The combination of this data within the therapeutic field is significant and considered one of the most important foundations for advancing health and social improvement [19].

Studies were also conducted to provide clinical epidemiological data from EB disease registries in Australia and the Netherlands, focusing on data collection in both administrative and clinical sections [13, 34].

Considering the importance of these cases and the lack of a MDS for EB in the country, which can limit the understanding, diagnosis, management, and research of EB in Iran, it

seems necessary to develop a MDS for this disease [23]. Providing a local dataset for EB in Iran, which ensures access to accurate and unambiguous data, offers a standard tool to facilitate communication between individuals and organizations involved in surveillance. It will also enable the comparison and analysis of activities by developing documentation methods.

Therefore, the aims of this study were to develop and validate the Iranian MDS for EB. In this regard, by systematically reviewing the scientific literature, we will identify essential data elements. Using the Delphi decision-making technique, subject matter experts will determine the degree of necessity of these data elements.

## 2. Material and methods

Several methods can be used to determine and develop a MDS, such as interviewing experts in each area, reviewing literature and documents, and examining available systems and datasets. This research is a descriptive cross-sectional study performed in 2023. A literature review and expert consensus were used to retrieve relevant data resources.

The EB minimum data set was developed through a three-step process, as outlined below: comprehensive literature review, classification of data elements, and validation of data elements using the Delphi technique.

### 2.1. Step I: Comprehensive literature review

The present study adopted the framework of the PRISMA guidelines to conduct a systematic review [35]. This systematic review was conducted using the PubMed, Scopus, and Web of Science databases without any time limitation until June 28, 2022. The websites of the National Health Service (NHS), the American Academy of Dermatology (AAD), the National Organization for Rare Disorders (NORD), and Debra of America were also searched. A keyword search of these databases was performed using terms related to minimum data set concepts (minimum data set, dataset, common data elements, data elements, data recording, data utilization, common data, data collection, national data set, core data set) and keywords related to epidermolysis bullosa (Epidermolysis Bullosa, Acantholysis Bullosa, Epidermolysis Bullosa in children). The search strategy was designed by two researchers [AS & SP] by combining two groups of keywords: "minimum dataset" and "epidermolysis bullosa" (Table 1). The keywords from the first group were retrieved from all fields, while the keywords from the second group were retrieved from the title and abstract. Based on the following inclusion and exclusion criteria, a decision was made regarding the inclusion of studies in this systematic review.

Inclusion criteria included all articles published in English that focused on establishing an EB disease registry and developing a minimum dataset for epidermolysis bullosa. Exclusion criteria were (1) conference abstracts, letters to the editor, dissertations or theses, and review articles or meta-analyses; (2) unavailability of full text for data extraction; (3) studies unrelated to the aim of the study; and (4) languages other than English.

The retrieved studies were imported into EndNote version 20 software. First, duplicates were identified and removed using the relevant software. Then, the titles and abstracts of all studies were evaluated according to the inclusion criteria, followed by an assessment of the full text of the articles to ensure that both inclusion and exclusion criteria were considered.

Two authors (HE & SP) manually screened the titles and abstracts of all articles to identify those relevant to the research objectives. Disagreements were resolved through discussion or consultation with a third reviewer (AS). The process of data extraction was carried out manually by a single reviewer (ShP) in Excel (Microsoft, 2019) and was subsequently verified by another reviewer (FA). Methodological quality was assessed independently by two reviewers using standardized tools for critical appraisal based on the Strengthening the Reporting of

**Table 1. Search strategies in three different databases.**

| Database | Search strategy syntax | Number of studies |
|---|---|---|
| **PubMed** | (("Minimum Data Set"[All fields] OR "Dataset" [All fields] OR "Common data elements" [All fields] OR "Data elements" [All fields] OR "Data recording" [All fields] OR "Data utilization" [All fields] OR "Common data" [All fields] OR "Data collection" [All fields] OR "national data set" [All fields] OR "Core data set" [All fields] OR "Dataset"[Mesh terms] OR "Common data elements" [Mesh terms] OR "Data collection" [Mesh terms]) AND ("Epidermolysis Bullosa" [Mesh terms] OR ("Epidermolysis Bullosa"[Title/Abstract] OR "Acantholysis Bullosa"[Title/Abstract] OR "Epidermolysis Bullosa children"[Title/Abstract])) | 400 |
| **Scopus** | (ALL ("Minimum Data Set") OR ALL (datasets) OR ALL ("common data elements") OR ALL ("data elements") OR ALL ("data recording") OR ALL ("data utilization") OR ALL ("common data") OR ALL ("data collection") OR ALL ("national data set") OR ALL ("core data set") AND TITLE-ABS-KEY ("epidermolysis bullosa") OR TITLE-ABS-KEY ("Epidermolysis Bullosa children") OR TITLE-ABS-KEY ("acantholytic bullosa")) | 53 |
| **Web of sciences** | ((ALL = ("Minimum Data Set") OR ALL = ("Dataset") OR ALL = ("Common data elements") OR ALL = ("Data elements") OR ALL = ("Data recording") OR ALL = ("Data utilization") OR ALL = ("Common data") OR ALL = ("Data collection") OR ALL = ("national data set") OR ALL = ("Core data set") OR ALL = ("Minimum Data Set") OR ALL = ("Dataset") OR ALL = ("Common data elements") OR ALL = ("Data elements") OR ALL = ("Data recording") OR ALL = ("Data utilization") OR ALL = ("Common data") OR ALL = ("Data collection") OR ALL = ("national data set") OR ALL = ("Core data set")) AND (TS = ("Epidermolysis Bullosa") OR TS = ("Acantholytic Bullosa") OR TS = ("Epidermolysis Bullosa children"))) | 69 |

Observational Studies in Epidemiology (STROBE) guidelines [36, 37]. The STROBE checklist was selected because the included studies were observational. The quality score was classified into six sections across three categories, from Category A to Category C. Any discrepancies were resolved through discussion and consensus among the authors.

## 2.2. Step II: Validation of data elements using the Delphi technique

Data elements were validated using the Delphi method in two rounds to reach a consensus on the data elements [38]. In the first round, the researchers designed a questionnaire. The measurement scale for the questions was Likert. The five-point Likert scale included "very important," "important," "moderately important," "slightly important," and "not important." The scoring method corresponded to 5, 4, 3, 2, and 1, respectively. A blank space was also provided at the end of each section for the experts to express their reasons and/or suggest modifications. The research sample in the first round was selected using purposive non-random sampling. Participants included dermatologists (n = 20) and health information management experts (n = 5), all with at least 5 years of experience and affiliated with universities of medical sciences (demographic information of the participants is presented in Fig 1). The sample size was determined to be 25 individuals based on the research team's opinion. The questionnaire was divided into two parts: an administrative section for health information management experts and a clinical section for dermatologists. The researcher-made questionnaire was distributed among the research population via e-mail or in person upon prior coordination.

The criteria used to determine the inclusion of data elements in the final MDS were based on the level of expert agreement. Data elements with an agreement level of over 75% were accepted in the first Delphi round, while those with an agreement level of 50% to 75% moved to the second round. Elements with an agreement level of less than 50% were removed in the

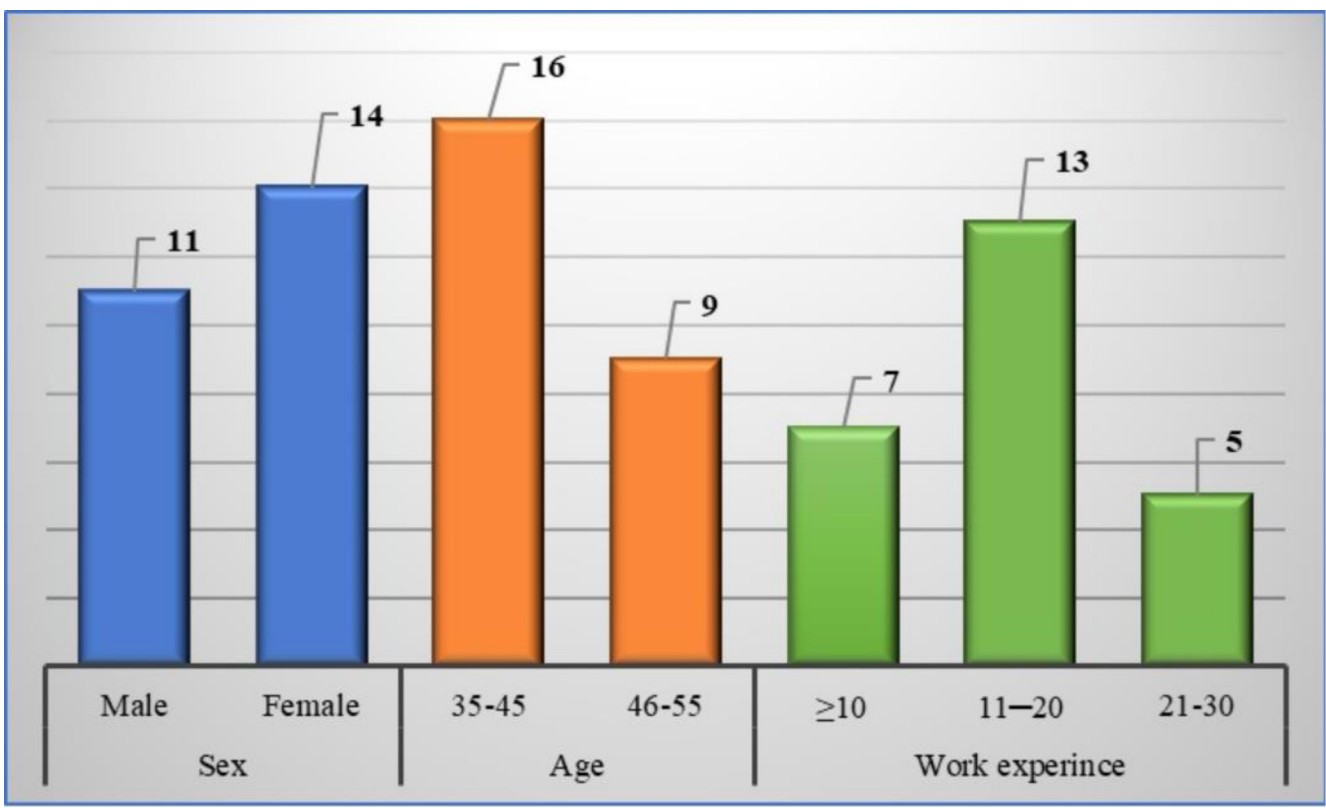

**Fig 1. Demographic information of the participants in the Delphi technique.**

first round. Items that received 50% to 75% approval from the experts in the first round, along with the suggested items, entered the second round for further review.

In the second round, the same questionnaire as that used in the first Delphi round was developed. Only elements with 50% to 75% agreement from the first round and any suggested by experts were included. As in the first round, elements that achieved 75% agreement from the experts were selected for inclusion in the MDS. The remaining elements were excluded. The participants in the second round of the Delphi technique were the same as those in the first round (five health information management experts and 20 dermatologists).

The final data elements for the MDS were determined through these two rounds, each lasting five weeks. The validity of the questionnaire was assessed through content validity. Six experts in health information management, medical informatics, and dermatology provided their opinions to confirm its relevance to the research. The reliability of the questionnaire was measured using Cronbach's alpha coefficient, which was found to be 0.837. For the Delphi study phase, data analysis was conducted using Microsoft Excel 2019, employing descriptive statistics such as numbers and frequency percentages.

## 2.3. Ethics approval and consent to participate

This study was approved by the Research Ethics Committee at Birjand University of Medical Sciences (Code: IR.BUMS.REC.1402.033). We obtained written informed consent from participants before their involvement in the focus group discussions and Delphi surveys. The consent form explicitly outlined the purpose of the study, its benefits, and the procedures for data

collection, storage, and analysis. No personally identifiable information was collected, and all data were anonymized prior to analysis.

## 3. Results

### 3.1. Comprehensive literature review

A total of 522 articles were retrieved from three databases. After removing duplicate articles (n = 27) and reviewing the titles and abstracts of the primary articles identified, 495 articles were selected for final review. After applying the inclusion and exclusion criteria of the study, 16 of these articles were considered for data extraction (For details see S1 Table). Data extracted from the websites of the National Health Service (NHS) [39], The American Academy of Dermatology (AAD) Association [40], National Organization for Rare Disorders (NORD) [41] and Debra of American [42] were also included in the study. Details of the search for articles are presented in Fig 2.

The oldest and most recent studies were conducted in 1991 and 2021, respectively (Fig 3). Most of the studies were conducted in the United States (n = 10, 62%). The remaining studies were conducted in Australia (n = 2, 13%), England (n = 2, 13%), Norway (n = 1, 6%), and the Netherlands (n = 1, 6%).

### 3.2. Classification of data elements

In the focus group meeting, eight sections were identified, which were classified into two categories: Administrative Data and Clinical Data. Administrative Data included 11 data elements in two sections: demographic data and residential address data. Clinical Data included 87 data elements across six sections: clinical symptoms, complications due to EB disease, diagnostic procedures, treatment procedures, death due to EB disease, and personal and family medical history.

### 3.3. Validation of data elements using the Delphi technique

Demographic information of the participants is presented in Fig 1. In the first Delphi round, 91 data elements were approved out of 98 items that could be assessed. Seven data elements (complete address of residence, esophagostenosis, bladder hypertrophy, chronic renal failure, constipation, vitamin D, and renal failure) received 50–75%.

Moreover, the experts suggested seven items (oral lesions, upper GI tract involvement, nasal mucosal lesions, airway obstruction, milia, pigmentation, and psychotherapy) as the items to be included in the second Delphi technique round. In the second round of the Delphi, of 14 elements that could be score, 12 items were confirmed and two data elements of chronic renal failure from the group of complications caused by EB disease and renal failure from the group of death caused by EB disease were rejected (Tables 2 and 3).

Finally, the MDS for EB disease was divided into administrative data and clinical data. The administrative data section was classified into two main categories, comprising 11 data elements. Of these 11 data elements, 5 were demographic data, and 4 pertained to residential address data.

The clinical data section was organized into seven main categories, with a total of 92 data elements. These data elements were distributed across the following sections: clinical symptoms (n = 25), complications caused by EB disease (n = 37), diagnostic procedures (n = 4), treatment procedures (n = 14), causes of death due to EB disease (n = 5), and personal and family medical history (n = 7). In total, the MDS for EB disease included 103 data elements. The classification of these data elements is presented in Table 4. (For details see S2 Table).

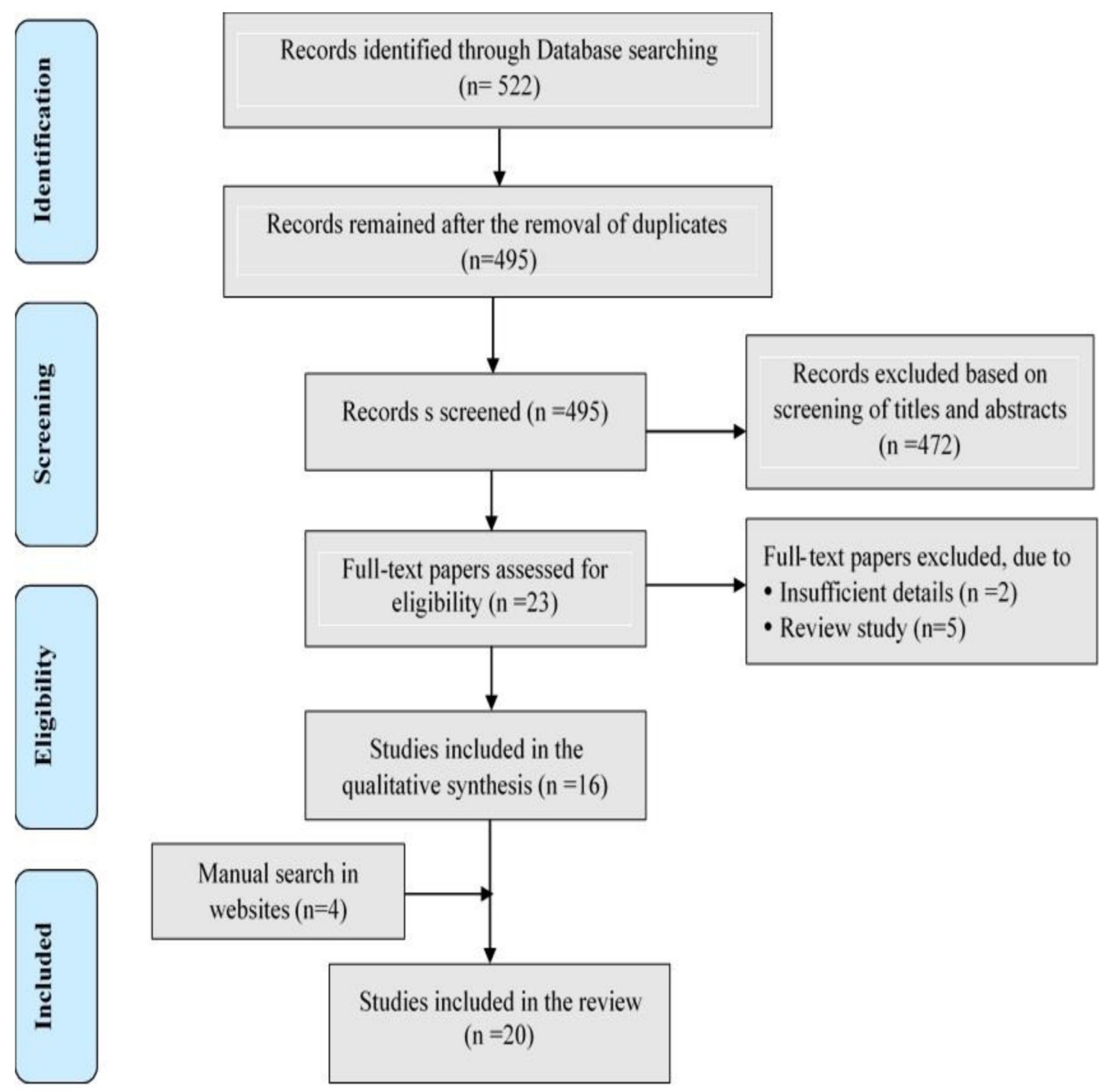

**Fig 2. The systematic review flowchart based on PRISMA protocol.**

As shown in Fig 4, the classification and number of elements in the clinical data category can be seen briefly. Most elements in this framework belong to complications caused by EB disease (n = 38, 40%), while the fewest are related to diagnostic procedures (n = 4, 4%).

## 4. Discussion

Given that the develop of MDS is an important prerequisite for collecting standard, integrated, and uniform data about diseases, the current research was conducted to develop MDS for

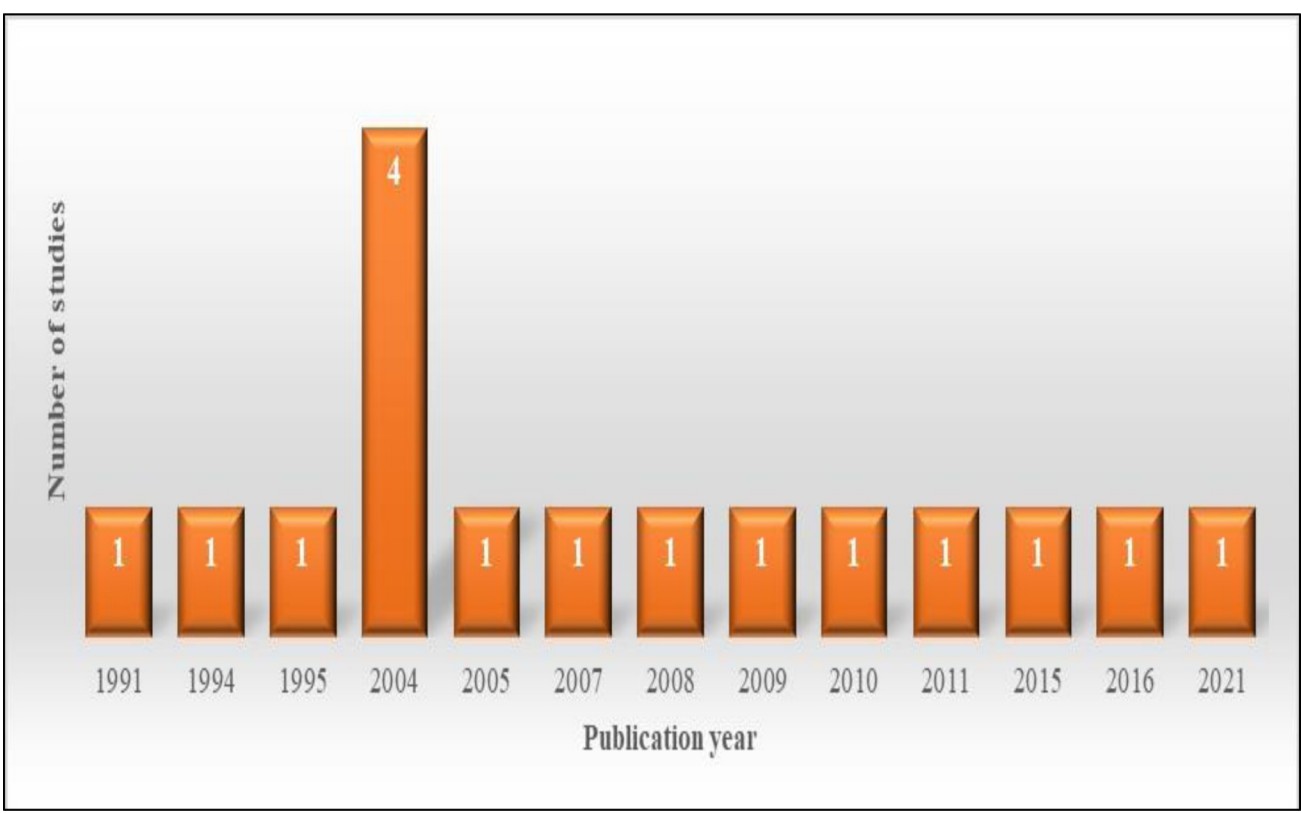

**Fig 3. The distribution of included studies based on publication years.**

epidermolysis bullosa. The increasing number of EB patients and limited financial resources in healthcare have made it necessary to establish an information management system or registry to fully collect all relevant data. This will help identify populations at risk, design programs for control and prevention, and promote health.

The results of the present study in the systematic review phase showed that most studies were conducted in the United States. Tang's study, which aimed to review the disease burden in patients with recessive dystrophic epidermolysis bullosa (RDEB), reported an analysis of the United States (US) National Epidermolysis Bullosa (EB) Registry. This registry, funded and operated from 1986 to 2002, reported an RDEB incidence of 3.05 cases per one million live births and a prevalence of 1.35 cases per one million live births [43]. It's essential that future research includes more studies from low- and middle-income countries to provide a more global perspective on EB [44].

As shown in the results section, based on expert agreement and consensus, the MDS for EB disease was ultimately modified to include 103 data elements divided into two main sections: administrative and clinical. Given the significant disparity between administrative and clinical

**Table 2. Administrative data category for the MDS for EB, Delphi technique.**

| Data Sections | Number of Data Elements | First Round of Delphi | | | | Second Round of Delphi | | | Final Number of Data Elements |
|---|---|---|---|---|---|---|---|---|---|
| | | <50% | 50–75% | 75%< | Suggested Data Elements | <50% | 50–75% | 75%< | |
| Demographic data | 5 | 0 | 0 | 5 | None | 0 | 0 | 0 | 5 |
| Residential address data | 6 | 0 | 1 | 5 | 1 | 0 | 0 | 1 | 6 |

**Table 3. Clinical data category for the MDS for EB, Delphi technique.**

| Data Sections | Number of Data Elements | First Round of Delphi | | | | | Second Round of Delphi | | | | Final Number of Data Elements |
|---|---|---|---|---|---|---|---|---|---|---|---|
| | | <50% | 50–75% | 75%< | Rejected Data Elements | Suggested Data Elements | <50% | 50–75% | 75%< | Rejected | |
| Clinical symptoms | | | | | | | | | | | |
| Anatomical site | 11 | 0 | 0 | 11 | 0 | 0 | 0 | 0 | 11 | 0 | 11 |
| Type of EB | 4 | 0 | 0 | 4 | 0 | 0 | 0 | 0 | 4 | 0 | 4 |
| Skin manifestations | 6 | 0 | 0 | 6 | 0 | 0 | 0 | 0 | 6 | 0 | 6 |
| Disease severity | 3 | 0 | 0 | 3 | 0 | 0 | 0 | 0 | 3 | 0 | 3 |
| Age of disease onset | 1 | 0 | 0 | 1 | 0 | 0 | 0 | 0 | 1 | 0 | 1 |
| Complications caused by EB disease | | | | | | | | | | | |
| Eye | 9 | 0 | 0 | 9 | 0 | 0 | 0 | 0 | 0 | 0 | 9 |
| Gastrointestinal system | 2 | 0 | 1 | 1 | 0 | 3 | 0 | 0 | 3 | 0 | 4 |
| Genitourinary system | 10 | 0 | 3 | 7 | 0 | 3 | 1 | 0 | 2 | 1 | 9 |
| Respiratory system | 2 | 0 | 0 | 2 | 0 | 2 | 0 | 0 | 2 | 0 | 4 |
| Growth retardation | 1 | 0 | 0 | 1 | 0 | 0 | 0 | 0 | 0 | 0 | 1 |
| Anemia | 1 | 0 | 0 | 1 | 0 | 0 | 0 | 0 | 0 | 0 | 1 |
| Musculoskeletal system | 2 | 0 | 0 | 2 | 0 | 0 | 0 | 0 | 0 | 0 | 2 |
| Nervous system | 1 | 0 | 0 | 1 | 0 | 0 | 0 | 0 | 0 | 0 | 1 |
| Skin complications | 3 | 0 | 0 | 3 | 0 | 2 | 0 | 0 | 2 | 0 | 5 |
| Carcinoma | 1 | 0 | 0 | 1 | 0 | 0 | 0 | 0 | 0 | 0 | 1 |
| Data of diagnostic procedures | | | | | | | | | | | |
| Biopsy or sampling data | 3 | 0 | 0 | 3 | 0 | 0 | 0 | 0 | 0 | 0 | 3 |
| Molecular genetic testing | 1 | 0 | 0 | 1 | 0 | 0 | 0 | 0 | 0 | 0 | 1 |
| Data of treatment procedures | | | | | | | | | | | |
| Complementary and alternative medicines | 2 | 0 | 1 | 1 | 0 | 1 | 0 | 0 | 1 | 0 | 2 |
| Medicines | 4 | 0 | 0 | 4 | 0 | 0 | 0 | 0 | 0 | 0 | 4 |
| Palliative procedures | 4 | 0 | 0 | 4 | 0 | 0 | 0 | 0 | 0 | 0 | 4 |
| Rehabilitation procedures | 3 | 0 | 0 | 4 | 0 | 1 | 0 | 0 | 1 | 0 | 4 |
| Death data from EB disease | | | | | | | | | | | |
| Cause of death | 5 | 0 | 1 | 4 | 0 | 1 | 1 | 0 | 0 | 1 | 4 |
| Date of death | 1 | 0 | 0 | 1 | 0 | 0 | 0 | 0 | 0 | 0 | 1 |
| Personal and family medical history data | | | | | | | | | | | |
| History of the person's disease | 4 | 0 | 0 | 4 | 0 | 0 | 0 | 0 | 0 | 0 | 4 |
| Family history of EB disease | 3 | 0 | 0 | 3 | 0 | 0 | 0 | 0 | 0 | 0 | 3 |

data, numerous studies have organized data elements in this manner. For instance, in the research focused on developing and utilizing a poisoning registry, the data elements extracted from various sources were classified into two primary categories: administrative data and clinical data. Ninety-eight data elements within the administrative data category were further divided into three sections: general information, admission details, and discharge information. In the clinical data category, one hundred thirty-one data elements were categorized into five sections: clinical observation records, clinical assessment records, medical history records, diagnostic information, and treatment plan details [45].

Nevertheless, in various other studies, alternative classifications have been employed based on the characteristics of the diseases. For instance, in the research titled "development of a national consensus minimum data set for the diagnosis and treatment of oral cancer," the proposed framework was divided into six sections: management data with four axes, historical

**Table 4. Final administrative & clinical data elements for EB.**

| Main class | Subclass | Data elements |
|---|---|---|
| **Administrative data** | Demographic data | national code, gender, date of birth, ethnicity, race |
| | Address data of residence | province, city, village, full address of residence, telephone and mobile phone number |
| **Clinical data** | Clinical symptoms | Anatomical site involved (genitalia, anus, hand, feet, elbow, knee, lips, face, oral mucosa, eyes and neck) |
| | | Type of EB disease (simplex, junctional, dominant dystrophic, recessive dystrophic) |
| | | Skin manifestations (blister, erosion, crust, scar, milia, exuberant granulation tissue) |
| | | Disease severity (Localized, Intermediate, Severe) |
| | | Age of disease onset |
| | Complications caused by EB disease | Eye (corneal blister, corneal scar, corneal corrosion or abrasion, symblepharons, blepharitis, ectropion, blockage of tear ducts, reduced vision and blindness) |
| | | Gastrointestinal system (esophagostenosis, abnormality or tooth decay, oral lesions and upper GI tract involvement) |
| | | Genitourinary system (ureteral stricture, urinary retention, bladder hypertrophy, hydronephrosis, ureteral stricture, pyelonephritis, cystitis, acute renal failure and constipation) |
| | | Respiratory system (upper respiratory system failure, lower respiratory system failure, nasal mucosa lesions and airway obstruction) |
| | | Growth retardation |
| | | Anemia |
| | | Musculoskeletal system (deformity of hands, and legs) |
| | | Nervous system (muscular dystrophy) |
| | | Skin complications (alopecia, hair loss, nail dystrophy, milia, pigmentation) |
| | | Carcinoma and neoplasms |
| | Diagnostic procedures | Biopsy or sampling data (skin, Chorionic Villus Sampling, amniocentesis) |
| | | Molecular genetic testing |
| | Treatment procedures | Complementary and alternative drugs (iron, vitamin D) |
| | | Medicines (antibiotics, antidepressants, pain reliever, epilepsy drugs) |
| | | Palliative procedures (using protective and non-adhesive dressings, draining blisters, reducing skin friction, keeping the skin cool) |
| | | Rehabilitation procedures (physiotherapy, ergotherapy, hydrotherapy, psychotherapy) |
| | Death data from EB disease | Cause of death from EB disease (death from sepsis, failure to thrive, airway obstruction, and squamous cell carcinoma) |
| | | Date of death |
| | Personal and family medical history data | History of the person's disease (underlying disease, name of the underlying disease, duration of the underlying disease, treatments provided) |
| | | Family history of EB disease (family relationship, type of disease, history of infection) |

data with four axes, paraclinical indicators with two axes, clinical indicators, data about therapeutic measures, and mortality statistics [46].

In 1986, the US National Institutes of Health launched the National EB Registry to better characterize this disease at the epidemiologic, clinical, ultrastructural, and molecular levels [47, 48]. The data elements in this registry are divided into two main sections. The demographic information section includes gender, race, ethnicity, age, past and present medical records, family history, and socio-economic parameters. In contrast, the clinical information section contains physical examination findings and diagnostic laboratory studies. Clinical symptoms include types of EB, skin manifestations (desquamation, scarring, severe granulomatous tissue, milia), complications caused by the disease (carcinoma, renal complications, eye complications, digestive complications, respiratory complications, musculoskeletal complications, growth retardation), causes of death (sepsis, non-infectious respiratory failure, carcinoma, and pneumonia), and age at death [47]. In the current study, the data set is defined in two main

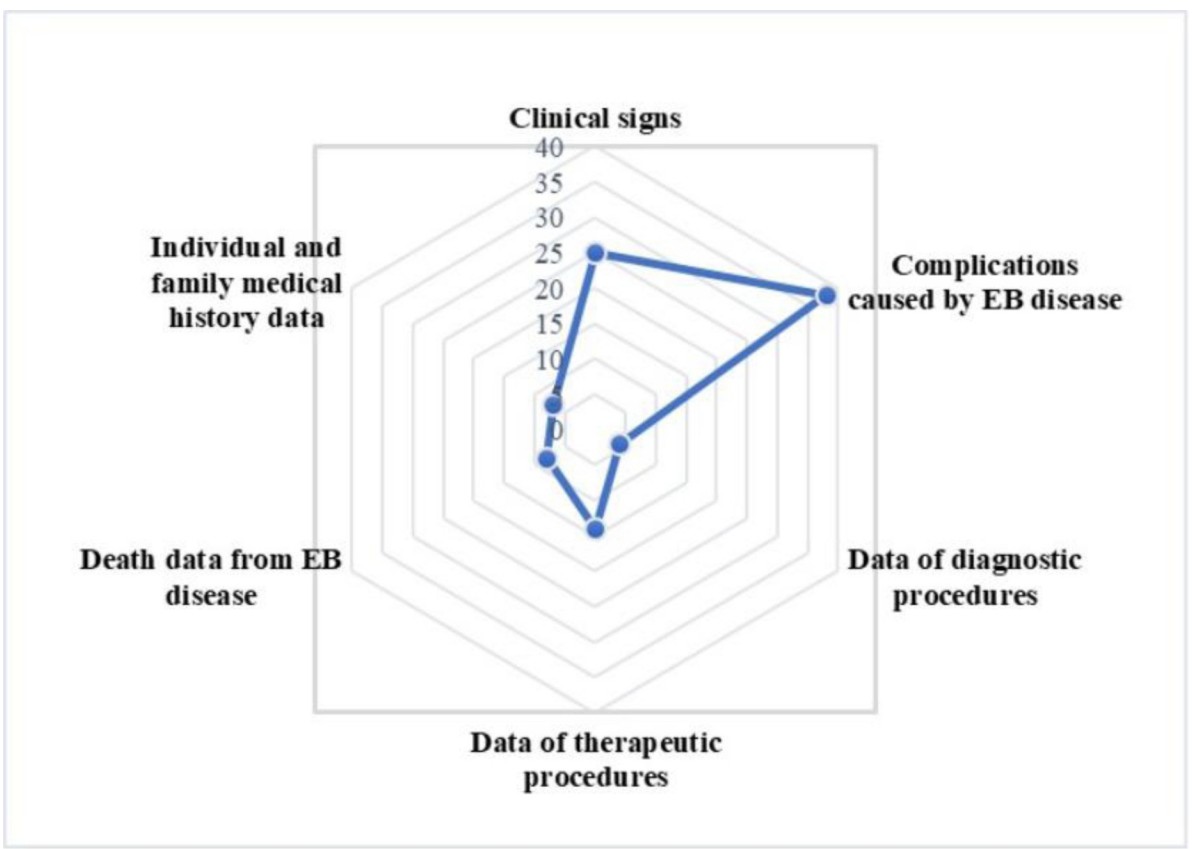

**Fig 4. The RADAR diagram of dispersion of data elements in the EB disease information system framework.**

classes: administrative and clinical. Regarding renal failure, which was present in the American registry but excluded from the data collection of this study, the reason for this difference is that kidney or renal involvement is rare in EB patients but usually occurs in those with severe forms. Therefore, from the experts' perspective, it was considered less important and was removed from the MDS [43].

In the study by C. Has et al., which aimed at the practical management of epidermolysis bullosa, clinical data elements included types of EB, anatomical complications (such as blisters and skin wounds, pain and itching, oral lesions, and tooth decay), diagnostic methods (skin biopsy and genetic testing), drug treatments (antibiotics), causes of death due to the disease (sepsis, lack of growth, severe anemia, and cancer), family history, and laboratory findings [49]. In this study, within the group of diagnostic procedures, in addition to skin biopsy, placental villi of the fetus and amniocentesis were also discussed and agreed upon by experts. Placental villi and amniocentesis are important in EB because they can be used to diagnose this disease during pregnancy and to identify the genetic mutations that cause it [50].

Additionally, in the drug data, in addition to the antibiotics mentioned in the above study, antidepressants, pain suppressants, and epilepsy medications were also identified for these patients. These drugs are important in the treatment of EB because they can help improve the quality of life for patients with this disease [51]. Since depression and anxiety are common in patients with EB and can affect their physical and mental health as well as their adherence to treatment, experts considered antidepressants to be important in the category of drug information. Furthermore, because EB patients experience pain caused by blisters, wounds, infections,

and surgery, the use of pain suppressants helps to enhance their quality of life [5]. In addition, in the current study, experts considered epilepsy medications to be important in the medication information category for EB patients because these drugs can help prevent or treat seizures that may occur in some forms of EB [52].

Yong CK et al., in a study aimed at providing clinical epidemiological data from the EB disease registry in Australia, developed the MDS according to the present study. The MDS was classified into two groups: administrative data, which includes the demographic characteristics of the patient (age, sex, ethnicity, and geographic distribution), and clinical data. Clinical data includes the type of disease, past and current medical history, family history, diagnostic laboratory studies, blister or erosion areas (legs and thighs, arms and forearms, inside the mouth, abdomen, fingers, armpits, face, neck, trunk, hands, elbows, cornea, scalp), complications caused by the disease (including complications of annular trachea; gastrointestinal complications; weight growth; findings related to hair, teeth, and nails; eye complications; and complications of the reproductive system), and causes of death due to the disease [34].

In relation to management data, race has been considered in some studies; however, this data item has been removed in Iran due to the lack of racial discrimination [31, 34, 47, 53]. Given that, in EB disease, family history is important in addition to individual medical history, it was also agreed upon by the experts' panel that family history should be included, as it is accepted in other similar registries. In the current study, the anatomical location involved in the genital organs was also proposed and agreed upon. Additionally, regarding the data on complications caused by EB disease, the musculoskeletal system, nervous system, carcinoma, and anemia were identified as elements that were not mentioned in the previous study among the clinical class.

In the study by Baardman & Bolling, which aimed to evaluate the epidemiological data of EB in the Netherlands using the EB patient registry, administrative data elements included demographic information (age, race, sex) and clinical data, such as types of EB, age at death, causes of death due to the disease, complications of the nervous system (muscular dystrophy), diagnostic measures (genetic testing), skin manifestations (blisters, mechanical fragility), and medical records of patients [13]. In the current study, all the data presented in the above study received the necessary points from the experts. In addition to the above, palliative and rehabilitation measures were also identified as part of the treatment data. According to the findings of the present study, palliative and rehabilitation measures are among the basic interventions for EB patients. While these measures were not mentioned in previous studies, they received necessary points from experts. Therefore, they were classified in the clinical data category under the treatment subsection.

It seems that several data elements in some of the above registries are not in accordance with the purposes of registry development. In other words, they can be obtained from the hospital information system (HIS). Additionally, the existence of a large amount of data can cause confusion and waste time; therefore, the researchers approved the afore mentioned classes and subclasses to develop the MDS, which was validated by experts.

### 4.1. Strengthen and limitation

The MDS for EB did not exist in Iran until now. The standard developed in this study allows for the global comparison of Iranian data. The quality of the designed MDS lies in its development based on recognized publications through a systematic review, rather than a preliminary review of the literature. As a clinical instrument, this MDS is fundamental not only to encourage EB registry design but also to assist in the collection of high-value clinical information essential for building a modern and superior clinical knowledge repository.

The primary function of the MDS is to deliver high-quality data to healthcare professionals to improve their care administration processes and facilitate informed decision-making. However, it is vital to recognize some limitations of this study. First, restricting our literature search to English may have led us to overlook numerous registries in other countries. Some countries may not provide information on the presentation, structure, and evaluation of their EB registries and databases, especially in low- and middle-income settings. Consequently, these were excluded from our investigation. The lack of extensive research on EB registries was one of the main limitations of this study.

## 5. Conclusion

Considering the increase in the number of EB patients in Iran, it seems necessary to establish a registry for EB patients in order to manage their care effectively. The first step in creating a disease registry is to identify information elements as part of an integrated and comprehensive framework, along with a suitable data dictionary. The use of standard elements in the MDS leads to improved understanding and interpretation of data. The MDS identified in this study, with the approval of experts, can be an effective step toward integrating the information of these patients in Iran and provides valuable approaches to improving the management of their information. Moreover, during their research, the researchers did not find clear guidelines for the development of registries. Given the increasing number of registries being developed, it is suggested that health researchers and policymakers focus on preparing these guidelines in future research.

## Supporting information

**S1 Table. PRISMA checklist for selected articles.**
(DOCX)

**S2 Table. Final minimum data set for EB (Frequency and percentage).**
(DOCX)

## Author Contributions

**Conceptualization:** Somayeh Paydar, Shahrbanoo Pahlevanynejad, Farkhondeh Asadi, Hamideh Ehtesham, Azam Sabahi.

**Data curation:** Somayeh Paydar, Azam Sabahi.

**Formal analysis:** Somayeh Paydar, Shahrbanoo Pahlevanynejad, Azam Sabahi.

**Investigation:** Azam Sabahi.

**Methodology:** Somayeh Paydar, Azam Sabahi.

**Project administration:** Azam Sabahi.

**Supervision:** Azam Sabahi.

**Validation:** Somayeh Paydar, Shahrbanoo Pahlevanynejad, Azam Sabahi.

**Visualization:** Shahrbanoo Pahlevanynejad.

**Writing – original draft:** Somayeh Paydar, Shahrbanoo Pahlevanynejad, Farkhondeh Asadi, Hamideh Ehtesham, Azam Sabahi.

**Writing – review & editing:** Somayeh Paydar, Shahrbanoo Pahlevanynejad, Farkhondeh Asadi, Hamideh Ehtesham, Azam Sabahi.

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
