## [Decision Letter · Decision Letter 0]

11 Oct 2024

PONE-D-23-32911Development and validation of the Iranian Minimum Data Set for Epidermolysis Bullosa: A mixed method approachPLOS ONE

Dear Dr. Sabahi,

Thank you for submitting your manuscript to PLOS ONE. After careful consideration, we feel that it has merit but does not fully meet PLOS ONE’s publication criteria as it currently stands. Therefore, we invite you to submit a revised version of the manuscript that addresses the points raised during the review process.

We look forward to receiving your revised manuscript.

Kind regards,

Bibi Razieh Hosseini Farash

Academic Editor

PLOS ONE

Journal Requirements:

3. We note that your Data Availability Statement is currently as follows: “All relevant data are within the manuscript and in Supporting Information files.”

Please confirm at this time whether or not your submission contains all raw data required to replicate the results of your study. Authors must share the “minimal data set” for their submission. PLOS defines the minimal data set to consist of the data required to replicate all study findings reported in the article, as well as related metadata and methods (https://journals.plos.org/plosone/s/data-availability#loc-minimal-data-set-definition). For example, authors should submit the following data: - The values behind the means, standard deviations and other measures reported; - The values used to build graphs; - The points extracted from images for analysis. Authors do not need to submit their entire data set if only a portion of the data was used in the reported study. If your submission does not contain these data, please either upload them as Supporting Information files or deposit them to a stable, public repository and provide us with the relevant URLs, DOIs, or accession numbers. For a list of recommended repositories, please see https://journals.plos.org/plosone/s/recommended-repositories. If there are ethical or legal restrictions on sharing a de-identified data set, please explain them in detail (e.g., data contain potentially sensitive information, data are owned by a third-party organization, etc.) and who has imposed them (e.g., an ethics committee). Please also provide contact information for a data access committee, ethics committee, or other institutional body to which data requests may be sent. If data are owned by a third party, please indicate how others may request data access.

Additional Editor Comments:

**Major Issues:**

**Lack of Clarity in Methods**:

The manuscript states the use of the Delphi technique, but the detailed methodology lacks clarity. There is no comprehensive explanation of the number of participants in each Delphi round, nor a clear rationale for why two rounds were sufficient. This leaves questions about the robustness of the Delphi process used.

**Limited Generalizability**:

While the study is focused on developing a Minimum Data Set (MDS) for Epidermolysis Bullosa in Iran, the manuscript does not discuss how the findings can be generalized or applied beyond Iran. Considering PLOS ONE's international audience, it is critical to emphasize how the developed MDS may have broader applications.

**Inconsistent Structure**:

There are several instances of structural inconsistency in the presentation of sections. For instance, the transitions between the "Methods" and "Results" sections are not smooth. The "Discussion" could have linked the results better with the broader global context and existing literature.

**Insufficient Statistical Analysis**:

The manuscript does not provide sufficient detail on the statistical analysis used to validate the data set. While descriptive statistics are mentioned, there is no mention of more robust statistical methods to ensure the reliability and validity of the identified data elements.

**Ethical Considerations**:

The ethical section mentions consent and ethical approval but lacks detail on how data confidentiality and privacy were maintained during the focus group discussions and Delphi surveys.

**Minor Issues:**

**Language and Grammar**:

The manuscript contains several minor grammatical errors and awkward sentence structures that can hinder comprehension. For example, the abstract and introduction sections have some overly long sentences that can be difficult to follow.

**Figures and Tables**:

The presentation of figures and tables is somewhat unclear. For example, Figure 2's flowchart is not described in detail, and Table 1 lacks clear labels and explanations. Improving their clarity would enhance the readability of the manuscript.

**References and Citations**:

While the references seem to be comprehensive, there is an over-reliance on older studies in the discussion section. Including more recent studies would strengthen the manuscript and make it more relevant to current research.

**Recommendations for Improvement:**

Clarify the methodology, particularly the Delphi technique, and provide detailed participant numbers and the rationale for the rounds.Discuss the potential international relevance of the Iranian MDS and how it can inform global standards.Improve the flow between sections, particularly by connecting the findings to broader literature in the discussion section.Ensure that the ethical considerations are more thoroughly explained, especially in regard to data privacy and confidentiality.Revise the manuscript for language clarity and structure, and improve the presentation of figures and tables.

Reviewers' comments:

Reviewer's Responses to Questions

**Comments to the Author**

1. Is the manuscript technically sound, and do the data support the conclusions?

Reviewer #1: Yes

2. Has the statistical analysis been performed appropriately and rigorously? 

Reviewer #1: Yes

3. Have the authors made all data underlying the findings in their manuscript fully available?

Reviewer #1: Yes

4. Is the manuscript presented in an intelligible fashion and written in standard English?

Reviewer #1: Yes

5. Review Comments to the Author

Reviewer #1: Comments

Let me congratulate the authors for taking the time to research such an important phenomenon. I find the following comments useful for enriching the content of the manuscript.

Abstract

• Under the abstract on line 33 page 1, the first statement is not clear.

Introduction

• The authors need to provide some background information on Epidermolysis bullosa in Iran

• The authors should also indicate the specific objectives of the study.

• An outline of the study should also be provided at the end of the introduction to give readers the direction of the study.

Methods

• Line 169 page 8, in case of disagreement, how many authors review the papers? The authors need to state the number of authors who review the papers in case the two researchers disagree.

• The sentence on lines 186 to 191 on page 9 is too long.

Results

• The authors should adopt the format (n=..., %) for their results presentation for lines 224 to 226 on page 10 as this (n=..., %) was done on page 16.

Discussion

• I suggest the authors give a title such as “Strengths and limitations of the study”.

• Apart from the validation of data elements using the Delphi method, has the minimum data set tool been tested? If not, can that be captured under the limitation?

6. PLOS authors have the option to publish the peer review history of their article (what does this mean?). If published, this will include your full peer review and any attached files.

Reviewer #1: No

---

## [Author Response · Author response to Decision Letter 0]

5 Dec 2024

Dear Editor;

Thank you very much for considering our manuscript and the opportunity for revision. We would like to express our appreciation for the constructive and valuable comments and suggestions offered by the reviewers. We considered all issues mentioned in the reviewers' comments carefully to improve the quality of our manuscript. All modifications are highlighted in yellow in the revised manuscript. 

Thank you for updating your data availability statement. You note that your data are available within the Supporting Information files, but no such files have been included with your submission. At this time we ask that you please upload your minimal data set as a Supporting Information file, or to a public repository such as Figshare or Dryad.

Authors’ Answer: 

Minimal data set added as Supporting Information files and cited in manuscript.

Major Issues:

1. Lack of Clarity in Methods:

The manuscript states the use of the Delphi technique, but the detailed methodology lacks clarity. There is no comprehensive explanation of the number of participants in each Delphi round, nor a clear rationale for why two rounds were sufficient. This leaves questions about the robustness of the Delphi process used. 

Authors’ Answer:

To enhance the clarity of the methodology, we will provide a more detailed explanation of the Delphi technique, the number of participants involved in each round of the Delphi technique and the reason for the two rounds in Step III (Validation of data elements using the Delphi technique) of section method.

2. Limited Generalizability:

While the study is focused on developing a Minimum Data Set (MDS) for Epidermolysis Bullosa in Iran, the manuscript does not discuss how the findings can be generalized or applied beyond Iran. Considering PLOS ONE's international audience, it is critical to emphasize how the developed MDS may have broader applications. 

Authors’ Answer:

The generalizability dimension of the study is the possibility of Iran's presence in comparing endemic disease data between different countries, which was added to the introduction according to the opinion of the respected referee.

3. Inconsistent Structure:

There are several instances of structural inconsistency in the presentation of sections. For instance, the transitions between the "Methods" and "Results" sections are not smooth. The "Discussion" could have linked the results better with the broader global context and existing literature.

Authors’ Answer:

The method was edited and presented based on three phases and the results are presented in the order of these three phases. Discussion was revised based on results. 

4. Insufficient Statistical Analysis:

The manuscript does not provide sufficient detail on the statistical analysis used to validate the data set. While descriptive statistics are mentioned, there is no mention of more robust statistical methods to ensure the reliability and validity of the identified data elements. 

Authors’ Answer:

We have explained both in the method section and in the findings section that the validity of the data was based on the Delphi technique. In this part, the percentage and frequency were important for us to decide whether the data elements should be approved, rejected, or asked again in the second stage.

5. Ethical Considerations:

The ethical section mentions consent and ethical approval but lacks detail on how data confidentiality and privacy were maintained during the focus group discussions and Delphi surveys. 

Authors’ Answer:

Details of this section were added.

Minor Issues:

1. Language and Grammar:

The manuscript contains several minor grammatical errors and awkward sentence structures that can hinder comprehension. For example, the abstract and introduction sections have some overly long sentences that can be difficult to follow.

Authors’ Answer:

According to the reviewer's opinion, the manuscript was grammatically edited.

2. Figures and Tables:

The presentation of figures and tables is somewhat unclear. For example, Figure 2's flowchart is not described in detail, and Table 1 lacks clear labels and explanations. Improving their clarity would enhance the readability of the manuscript.

Authors’ Answer:

We improved the clarity of figures and tables. Details of Fig 2 explained in result section (3.1. Comprehensive literature review). Details of Table 1 explained in method section (Comprehensive literature review).

3. References and Citations:

While the references seem to be comprehensive, there is an over-reliance on older studies in the discussion section. Including more recent studies would strengthen the manuscript and make it more relevant to current research. 

Authors’ Answer:

According to the reviewer's opinion, references reviewed.

Reviewer 1 

Abstract

• Under the abstract on line 33 page 1, the first statement is not clear.

Authors’ Answer:

According to the reviewer's opinion, the desired corrections were made.

Introduction

• The authors need to provide some background information on Epidermolysis bullosa in Iran 

Authors’ Answer:

According to the reviewer's opinion, background information on Epidermolysis bullosa in Iran was added to the introduction.

• The authors should also indicate the specific objectives of the study. 

Authors’ Answer:

According to the reviewer's opinion, the specific objectives of the study was added to the introduction

• An outline of the study should also be provided at the end of the introduction to give readers the direction of the study. 

Authors’ Answer:

According to the reviewer's opinion, an outline of the study was added to the end of the introduction

Methods

• Line 169 page 8, in case of disagreement, how many authors review the papers? The authors need to state the number of authors who review the papers in case the two researchers disagree.

Authors’ Answer:

The items mentioned in the method section were added. (line 163-167)

• The sentence on lines 186 to 191 on page 9 is too long. 

Authors’ Answer:

The sentence was revised.

Results

• The authors should adopt the format (n=..., %) for their results presentation for lines 224 to 226 on page 10 as this (n=..., %) was done on page 16. 

Authors’ Answer:

Mentioned format applied for all results.

Discussion

• I suggest the authors give a title such as “Strengths and limitations of the study”. 

Authors’ Answer:

This section is in the manuscript and was highlighted.

• Apart from the validation of data elements using the Delphi method, has the minimum data set tool been tested? If not, can that be captured under the limitation?

Authors’ Answer:

The data set was extracted through a systematic review of scientific literature and validated by subject matter experts using the Delphi decision-making technique. This is a standard method for determining MDS and is not a limitation. Several studies have been published using this method, for example: 

1. Development and validation of the Neonatal Abstinence Syndrome Minimum Data Set (NAS-MDS): a systematic review, focus group discussion, and Delphi technique.

2. Defining the content of a minimal dataset for acquired brain injury using a Delphi procedure.

---

## [Editor Report · Decision Letter 1]

17 Dec 2024

Development and validation of the Iranian Minimum Data Set for Epidermolysis Bullosa: A mixed method approach

PONE-D-23-32911R1

Dear Dr. Azam Sabahi,

We’re pleased to inform you that your manuscript has been judged scientifically suitable for publication and will be formally accepted for publication once it meets all outstanding technical requirements.

Kind regards,

Bibi Razieh Hosseini Farash

Academic Editor

PLOS ONE

Additional Editor Comments (optional):

**Abstract and Introduction:<o:p></o:p>**

The abstract is well-structured, covering all key components of the study.<o:p></o:p>

The introduction clearly outlines the research problem and the significance of the study.<o:p></o:p>

**Methodology:<o:p></o:p>**

The methodology section provides a detailed description of the processes, including the use of the Delphi technique and systematic review.<o:p></o:p>

Data selection criteria and inclusion/exclusion parameters are well-defined.<o:p></o:p>

**Results:<o:p></o:p>**

Tables and figures are clearly organized, and changes based on reviewer comments have been incorporated.<o:p></o:p>

Data categorization into administrative and clinical data aligns well with the study's objectives.<o:p></o:p>

**Discussion and Conclusion:<o:p></o:p>**

The discussion effectively analyzes the findings and compares them with previous studies.<o:p></o:p>

The conclusion is concise, clear, and aligned with the research objectives.<o:p></o:p>

**References:<o:p></o:p>**

The references are up-to-date, relevant, and properly formatted.<o:p></o:p>
---

## [Editor Report · Acceptance letter]

26 Dec 2024

PONE-D-23-32911R1 

PLOS ONE

Dear Dr. Sabahi, 

I'm pleased to inform you that your manuscript has been deemed suitable for publication in PLOS ONE. Congratulations! Your manuscript is now being handed over to our production team.

Kind regards, 

on behalf of

Dr. Bibi Razieh Hosseini Farash 

Academic Editor

PLOS ONE